# SMTP-44D Exerts Antioxidant and Anti-Inflammatory Effects through Its Soluble Epoxide Hydrolase Inhibitory Action in Immortalized Mouse Schwann Cells upon High Glucose Treatment

**DOI:** 10.3390/ijms23095187

**Published:** 2022-05-06

**Authors:** Ryosuke Shinouchi, Keita Shibata, Shiori Jono, Keiji Hasumi, Koji Nobe

**Affiliations:** 1Division of Pharmacology, Department of Pharmacology, Toxicology & Therapeutics, School of Pharmacy, Pharmacology Research Center, Showa University, 1-5-8 Hatanodai, Shinagawa-ku, Tokyo 142-8555, Japan; kshibata@pharm.showa-u.ac.jp (K.S.); 13116100p@stud.showa-u.ac.jp (S.J.); 2Department of Applied Biological Science, Tokyo University of Agriculture and Technology, 3-5-8 Saiwaicho, Fuchu-shi, Tokyo 183-8509, Japan; hasumi@cc.tuat.ac.jp; 3Division of Research and Development, TMS Co., Ltd., 1-23-3-501 Miyamachi, Fuchu-shi, Tokyo 183-0023, Japan

**Keywords:** diabetic neuropathy, IMS32, Schwann cell, soluble epoxide hydrolase, epoxyeicosatrienoic acid, antioxidant, anti-inflammatory, SMTP, SMTP-44D

## Abstract

Diabetic neuropathy (DN) is a major complication of diabetes mellitus. We have previously reported the efficacy of *Stachybotrys microspora* triprenyl phenol-44D (SMTP-44D) for DN through its potential antioxidant and anti-inflammatory activities. However, the mechanisms underlying the antioxidant and anti-inflammatory activities of SMTP-44D remain unclear. The present study aimed to explore the mechanism of these effects of SMTP-44D in regard to its inhibition of soluble epoxide hydrolase (sEH) in immortalized mouse Schwann cells (IMS32) following high glucose treatment. IMS32 cells were incubated in a high glucose medium for 48 h and then treated with SMTP-44D for 48 h. After incubation, the ratio of epoxyeicosatrienoic acids (EETs) to dihydroxyeicosatrienoic acids (DHETs), oxidative stress markers, such as NADPH oxidase-1 and malondialdehyde, inflammatory factors, such as the ratio of nuclear to cytosolic levels of NF-κB and the levels of IL-6, MCP-1, MMP-9, the receptor for the advanced glycation end product (RAGE), and apoptosis, were evaluated. SMTP-44D treatment considerably increased the ratio of EETs to DHETs and mitigated oxidative stress, inflammation, RAGE induction, and apoptosis after high glucose treatment. In conclusion, SMTP-44D can suppress the induction of apoptosis by exerting antioxidant and anti-inflammatory effects, possibly through sEH inhibition. SMTP-44D can be a potential therapeutic agent against DN.

## 1. Introduction

Diabetic neuropathy (DN), along with retinopathy and nephropathy, is one of the most frequent diabetic complications, affecting approximately 50% or more of patients with diabetes [1]. DN appears in the early stages of diabetes, with symptoms of allodynia and hyperalgesia. The pathological progression of DN may lead to paresthesia and, in the worst case, gangrene leading to amputation of the foot [2]. Therefore, DN has become a social problem that severely degrades the quality of life and causes a massive increase in medical costs. The persistence of hyperglycemia induces oxidative stress and fatty acid increases, which activate the DN-associated metabolic pathways, namely the protein kinase C, polyol, advanced glycation end products, and hexosamine pathways. Especially, oxidative stress is considered the final common pathway of cellular injury under hyperglycemic conditions [3]. Furthermore, not only oxidative stress but also inflammation plays an important role in structural and functional damage associated with the pathologic progression of DN [4,5,6]. The underlying etiology of DN is multifactorial, and multiple pathways are involved in DN pathogenesis. However, detailed mechanisms underlying DN have not yet been elucidated.

*Stachybotrys microspora* triprenyl phenols (SMTPs) are a family of small-molecule triprenyl phenol metabolites derived from the fungus *S. microspora* [7,8]. Among the SMTPs, SMTP-7 was identified to show excellent therapeutic activities in several types of ischemic models in rodents and monkeys [9,10,11,12,13,14,15]. On the other hand, SMTP-44D [16] has been reported to have effective antioxidant and anti-inflammatory activities [17,18,19]. Recently, we demonstrated that SMTP-44D improves neural function, mechanical allodynia, and thermal hyperalgesia associated with DN through its antioxidant and anti-inflammatory activities [20]. However, we have not yet clarified the underlying mechanisms by which SMTP-44D improves neurological function via its antioxidant and anti-inflammatory activities.

It is postulated that SMTP-44D inhibits inflammation by inhibiting soluble epoxide hydrolase (sEH) [8,17]. Moreover, SMTP-44D exhibits antioxidative activity owing to its chemical structure [8,18]. sEH hydrolyzes epoxyeicosatrienoic acids (EETs) to dihydroxyeicosatrienoic acids (DHETs). EETs are potent endogenous signaling molecules involved in anti-inflammatory, vascular dilation, angiogenesis, neuroprotection, and analgesia [21,22,23]. This suggests that the antioxidant and anti-inflammatory activities of SMTP-44D via an sEH inhibition may improve DN-related neural function, mechanical allodynia, and thermal hyperalgesia.

Schwann cells are glial cells of the peripheral nervous system that support neurons and maintain the structural and functional integrity of nerves. In diabetes, Schwann cells are subjected to hyperglycemic insults, and their supporting functions are disrupted, resulting in peripheral nerve dysfunction [24,25]. The collapse of mitochondrial function in Schwann cells associated with glial support can cause primary neuronal degeneration, suggesting that Schwann cell dysfunction directly affects neural function [26]. Therefore, Schwann cell lines, such as immortalized mouse Schwann cells (IMS32), have been widely applied to in vitro models of DN [24,27].

Thus, the present study aimed to explore the mechanism underlying the antioxidant and anti-inflammatory effects of SMTP-44D via its sEH inhibitory action.

## 2. Results

### 2.1. Ratio of EETs to DHETs in Response to SMTP-44D in IMS32 under High Glucose Conditions

The ratio of 11(12)- and 14(15)-EETs to corresponding DHETs was measured to evaluate the sEH inhibition effect. Figure 1 shows the effects of SMTP-44D on the ratio of 11(12)- and 14(15)-EET/DHET as assessed by LC-ESI-MS at 96 h. In the high glucose, normal saline (HG + NS) group, the ratio of 11(12)-EET/DHET (*p* < 0.01, 0.36 ± 0.08) and 14(15)-EET/DHET (*p* < 0.05, 2.23 ± 0.38) was significantly decreased compared to that in the normal glucose, normal saline (NG + NS) group (1.72 ± 0.32 and 4.61 ± 0.81, respectively). The HG + SMTP-44D (30 μM) treatment group showed a significant increase in the ratio of 11(12)-EET/DHET (*p* < 0.01 1.34 ± 0.18) and 14(15)-EET/DHET (*p* < 0.05, 3.72 ± 0.55) compared to the HG + NS group.

### 2.2. Effects of SMTP-44D on NF-κB Nuclear Migration in IMS32 Cells under High Glucose Conditions

The levels of nuclear NF-κB, full-length and soluble forms of RAGE (f-RAGE and s-RAGE, respectively), and MMP-9, which cleaved f-RAGE to form sRAGE, were determined to evaluate the impact of SMTP-44D on high glucose-induced inflammatory responses in IMS32 cells. Figure 2 shows the effects of SMTP-44D on the NF-κB nuclear migration (nuclear fractions/nuclear fractions + cytosolic fractions) in the lysate, f-RAGE (RAGE in the lysate), s-RAGE (RAGE in the supernatant), and MMP-9 in the supernatant, as assessed by ELISA at 96 h. In the HG + NS group, nuclear migration of NF-κB (*p* < 0.01, 0.33 ± 0.06), f-RAGE (*p* < 0.01, 156.07 ± 16.08 pg/mg protein), s-RAGE (*p* < 0.01, 257.82 ± 79.21 pg/mg protein), and MMP-9 (*p* < 0.01, 0.20 ± 0.04 ng/mg protein) were significantly increased as compared to those in the NG + NS group (0.10 ± 0.02; 24.09 ± 6.63 pg/mg protein; 20.60 ± 2.25 pg/mg protein; and 0.06 ± 0.01 ng/mg protein, respectively). The HG + SMTP-44D (30 μM) treatment group resulted in significant decreases in the nuclear migration of NF-κB (*p* < 0.05, 0.14 ± 0.05), f-RAGE (*p* < 0.01, 65.53 ± 17.56 pg/mg protein), s-RAGE (*p* < 0.05, 43.74 ± 1.63 pg/mg protein), and MMP-9 (*p* < 0.05, 0.10 ± 0.004 ng/mg protein) compared to the HG + NS group.

### 2.3. Levels of NOX-1, MDA, IL-6, and MCP-1 in Response to SMTP-44D in IMS32 Cells under High Glucose Conditions

The levels of NOX-1, MDA as a measure of reactive oxygen species, IL-6, and MCP-1 were measured to evaluate the exacerbation of inflammation in IMS32 cells under high glucose conditions. Figure 3 summarizes the effects of SMTP-44D on the production of MDA in the lysate as assessed by the TBARS assay and on the expression of NOX-1 in the lysate as well as the production of IL-1β and IL-6 in the lysate as assessed by ELISA at 96 h. In the HG + NS group, the levels of NOX-1 (*p* < 0.01, 1.74 ± 0.22 ng/mg protein), MDA (*p* < 0.01, 58.9 ± 14.03 nmol/mg protein), IL-6 (*p* < 0.01, 49.02 ± 11.76 pg/mg protein), and MCP-1 (*p* < 0.01, 138.01 ± 19.04 ng/mg protein) were significantly increased as compared to those in the NG + NS group (0.29 ± 0.05 ng/mg protein; 2.86 ± 1.16 nmol/mg protein; 15.71 ± 3.34 pg/mg protein; and 54.80 ± 13.32 ng/mg protein, respectively). The HG + SMTP-44D (30 μM) treatment group showed significant decreases in the levels of NOX-1 (*p* < 0.01, 0.76 ± 0.04 ng/mg protein), MDA (*p* < 0.01, 11.29 ± 0.59 nmol/mg protein), IL-6 (*p* < 0.01, 9.66 ± 4.87 pg/mg protein), and MCP-1 (*p* < 0.01, 75.78 ± 2.09 ng/mg protein) compared to the HG + NS group.

### 2.4. Effects of SMTP-44D on Apoptosis in IMS32 Cells under High Glucose Conditions

TUNEL-positive cells were measured to detect Schwann cell apoptosis. Figure 4 shows the effects of SMTP-44D on apoptosis, as assessed by the TUNEL assay at 96 h. In the HG + NS group, the number of TUNEL-positive cells (*p* < 0.05, 1.66 ± 0.34%) was significantly higher than that in the NG + NS group (0.54 ± 0.16%). The HG + SMTP-44D (30 μM) treatment group showed a significant decrease in the number of TUNEL-positive cells (*p* < 0.05, 0.62 ± 0.35) compared to that in the HG + NS group.

## 3. Discussion

The present study suggested that SMTP-44D exhibited antioxidant and anti-inflammatory effects through the inhibition of sEH in IMS32 cells following high glucose treatment. We confirmed that SMTP-44D inhibited the metabolism of EETs to DHETs, which could be implicated in the reduced migration of NF-κB from the cytoplasm to the nucleus. The antioxidant effect of SMTP-44D may be related to the inhibition of the nuclear migration of NF-κB, which decreases the expression of the f-RAGE, leading to a decrease in the expression of NOX-1. This, in turn, suppresses the expression of reactive oxygen species as assessed as a decrease in MDA. The SMTP-44D-mediated reduction in NF-κB nuclear migration can be attributed to the suppression of IL-6, MCP-1, and MMP-9. Moreover, the decreased level of s-RAGE, which is produced by a cleavage of f-RAGE by MMP-9, can be a result of the decreased production of MMP-9. These antioxidant and anti-inflammatory effects of SMTP-44D may contribute to the mechanism of the suppression of Schwann cell apoptosis, which is involved in the pathological development of DN.

EETs are involved in the downregulation of NF-κB activation [28,29]. This fact suggests that the anti-inflammatory effect of EETs was exerted via the suppression of NF-κB activity (Figure 1). Node K et al. [30] demonstrated the anti-inflammatory effects of 11(12)-EET. Additionally, 14(15)-EET protects neurons from apoptosis by promoting mitochondrial biogenesis and function [31]. It is possible that SMTP-44D inhibits apoptosis not only by downregulating NF-κB activation but also by affecting the mitochondrial pathway. The NF-κB-dependent upregulation of f-RAGE expression leads to oxidative damage due to lipid peroxides and hydrogen peroxide [32]. In the present study (Figure 2), we confirmed that the high glucose treatment increased f-RAGE expression and nuclear migration of NF-κB, which have been implicated in the induction of several pro-inflammatory cytokines and chemokines, such as IL-6, MCP-1, and MMP-9 [33,34]. Indeed, we observed significant increases in IL-6, MCP-1, and MMP-9 secretion from IMS32 cells (Figure 2 and Figure 3). MMP-9 is reported to be involved in the cleavage of f-RAGE on the cell membrane and is released from the surface of the cell membrane as s-RAGE [35,36]. As shown in Figure 2, s-RAGE was abundant in the supernatant, along with a significant increase in MMP-9 secretion upon high glucose treatment. The binding of f-RAGE to advanced glycation end-product (AGE) activates NOX, which enhances intracellular oxidative stress and induces the secretion of various cytokines through NF-κB activation [37]. In the present study, f-RAGE expression was significantly increased by the high glucose treatment, suggesting that the increase in NOX-1 expression was accompanied by an increase in the release of reactive oxygen species, leading to an increase in MDA expression (Figure 2 and Figure 3). Schwann cell apoptosis is caused by high glucose-induced oxidative stress and inflammatory factors, which are involved in DN pathogenesis [38,39]. Furthermore, Schwann cell apoptosis has been reported to induce myelin degeneration [38]. In the present study, we suggest that apoptosis was induced by an increase in MDA, IL-6, and MCP-1 secretion in IMS32 cells after high glucose treatment (Figure 4). In addition, our previous report showed a thinning of myelin in the sciatic nerve caused by hyperglycemia [20]; therefore, myelin degeneration could be caused by apoptosis of Schwann cells in the sciatic nerve.

Since the main purpose of this study was to explore the mechanism of the action of SMTP-44D, a dose-dependent study was not performed. In addition, SMTP-44D could have ameliorated neuropathy by exerting effects in addition to the inhibition of sEH C-terminal domain epoxide hydrolase (C-EH) since SMTP-44D inhibits N-terminal domain phosphatase activity along with the C-EH [19,40]. Therefore, it is necessary to clarify the differences in activity between SMTP-44D and a highly selective inhibitor of the C-EH such as 12-(3-adamantan-1-yl-ureido) dodecanoic acid (AUDA).

In future in vitro studies, we will explore the dose-dependent effects of SMTP-44D on mitochondrial functions and compare the effects of SMTP-44D with those of AUDA. In vivo studies will confirm the apoptosis of Schwann cells in the sciatic nerve in a DN mouse model.

The present study is the first to show the antioxidant and anti-inflammatory effects of SMTP-44D in IMS32 cells through its sEH inhibitory effect. Our results suggest that EETs could ameliorate axonal damage by inhibiting apoptosis and maintaining the function of Schwann cells through their effects. Thus, SMTP-44D can serve as a new therapeutic agent for DN treatment due to its sEH inhibitory activity.

## 4. Materials and Methods

### 4.1. Reagents

Dulbecco’s modified Eagle’s medium (DMEM), trypsin-EDTA solution, and protease inhibitor cocktail (PIC) were purchased from Sigma-Aldrich Co., LLC. (St. Louis, MO, USA). RIPA buffer, leukotriene B4-d4 (LTB4-d4), (±)11(12)-EET (11(12)-EET), (±)11(12)-DiHET (11(12)-DHET), (±)14(15)-EET (14(15)-EET), and (±)14(15)-DiHET (14(15)-DHET) were purchased from Cayman Chemical Company (Ann Arbor, MI, USA). Four percent paraformaldehyde phosphate-buffered solution, trisodium citrate, formic acid (abt.99%) for HPLC, ultrapure water for LC/MS, 2-propanol for LC/MS, acetonitrile for LC/MS, and ethyl acetate were purchased from FUJIFILM Wako Pure Chemical Corporation (Osaka, Japan). Triton^®^ X-100 was purchased from MP BIOMEDICALS. (Santa Ana, CA, USA). Cellstain^®^-DAPI solution was purchased from Dojindo Laboratories (Kumamoto, Japan). SMTP-44D [9] was generously donated by TMS Co., Ltd. (Tokyo, Japan). The structure of SMTP-44D is shown in Figure 5.

### 4.2. Cell Culture

Immortalized mouse Schwann cells (IMS32) were purchased from Cosmo Bio Co. Ltd. (Tokyo, Japan). IMS32 were seeded on 100 mm/Tissue Culture Dish (AGC TECHNO GLASS Co., Ltd. [IWAKI], Shizuoka, Japan) or 9.5 mm × 4 wells Multi-Well Glass Bottom Dish (Matsunami Glass Ind., Ltd., Osaka, Japan) and cultured in culture medium for Schwann cell line IMS32 (Cosmo Bio Co. Ltd., Tokyo, Japan) at 37 °C under 5% CO_2_/95% air.

### 4.3. Treatment Protocol

When the cells reached approximately 90% confluency, the culture medium was replaced with serum-free DMEM supplemented with an N-2 supplement (Thermo Fisher Scientific K.K., Tokyo, Japan) containing normal glucose (5.6 mM; NG) or high glucose (30 mM; HG). IMS32 cells were incubated for 96 h under three different culture conditions: (1) NG + normal saline (NS) group; (2) NG + NS group; and (3) NG + SMTP-44D (30 μM) treatment group. The three different treatment conditions are shown in Figure 6.

### 4.4. Cell Sampling

After 96 h of incubation under each experimental condition, the cell culture supernatant was collected in ice-cold tubes. The cells were rinsed with phosphate-buffered saline (PBS; Takara Bio Inc., Shiga, Japan), mixed with 800 μL of RIPA buffer with PIC, detached from the 100 mm/Tissue Culture Dish by Cell Lifter (AS ONE Corporation, Osaka, Japan), and collected in ice-cold tubes. These cell lysates were homogenized and centrifuged at 10,000× *g* for 15 min at 4 °C; the supernatant was immediately dispensed in new ice-cold tubes. The samples were immediately frozen in liquid nitrogen and stored at −80 °C until they were examined for use in LC-EIS-MS, ELISA, or TBARS assay.

### 4.5. Measurement of EET and DHET by Liquid Chromatography-Electrospray Ionization Mass Spectrometry (LC-ESI-MS)

For the extraction of EET and DHET from the cell lysate, 100 μL of 0.1% formic acid containing ultrapure water was added to 500 μL of each sample. Then, the samples were incubated with ethyl acetate, vortexed, and centrifuged at 20,000× *g* for 5 min at 4 °C. The organic layer (upper layer) was then retrieved and evaporated to dryness using an evaporator. After the solvent evaporated to dryness, the EETs and DHETs were resuspended in 50 μL of mobile phase A (ultrapure water/acetonitrile/formic acid = 63/37/0.02) containing an internal standard (25 pg of LTB4-d4) and injected into the LC-ESI-MS system. All MS analyses were performed using a Prominence HPLC system (Shimadzu Corporation, Kyoto, Japan) equipped with a linear ion trap quadrupole mass spectrometer (QTRAP 5500, AB Sciex Pte. Ltd., Framingham, MA, USA), as previously described [41,42]. EETs and DHETs were subsequently analyzed using a tandem quadrupole mass spectrometer via multiple-reaction monitoring (MRM) in negative-ion mode. The m/z transitions monitored were as follows: 319.05/167.00 for 11(12)-EET, 337.10/167.20 for 11(12)-DHET, 319.06/219.00 for 14(15)-EET, 337.12/207.08 for 14(15)-DHET, and 339.13/196.93 for LTB4-d4. These EETs and DHETs were identified in the samples by matching their MRM signals and LC retention times with those of a pure standard. EETs and DHETs were quantified using standard curves of the analyte to the internal standard. EETs and DHETs were reported as a ratio.

### 4.6. Enzyme-Linked Immunosorbent Assay (ELISA)

The levels of full-length receptor for the advanced glycation end product (f-RAGE), monocyte chemotactic protein-1 (MCP-1), and interleukin-6 (IL-6) in the lysate and soluble RAGE (s-RAGE) and matrix metalloproteinase-9 (MMP-9) in the supernatant were assessed using an ELISA kit (R&D Systems Inc., Minneapolis, MN, USA) according to the manufacturer’s instructions. The levels of the oxidative stress indicator NADPH oxidase-1 (NOX-1) were assessed using the ELISA kit (MyBioSource Inc., San Diego, CA, USA) as described previously [15]. The NF-κB p65 ELISA kit (Abcam plc., Cambridge, UK) was used to determine the levels of nuclear factor-kappa B (NF-κB) in the cytosolic and nuclear fractions isolated from the IMS32 samples using a Nuclear/Cytosol Fractionation Kit (BioVision Inc., Milpitas, CA, USA). The total protein levels were quantitated using the PierceTM bicinchoninic acid protein assay kit (Thermo Fisher Scientific Inc., Coon Rapids, MN, USA) according to the manufacturer’s instructions. The levels of all proteins, except for NF-κB, were determined by interpolation from the standard curves and normalized for the protein content of each sample. Levels of NF-κB were quantified as the ratio of nuclear fractions to nuclear fractions + cytosolic fractions in the lysate.

### 4.7. Thiobarbituric Acid Reactive Substances (TBARS) Assay

The levels of oxidative stress in cell lysates were evaluated as an index of lipid peroxidation by measuring the levels of malondialdehyde (MDA) using the TBARS assay kit (Cayman Chemical, Ann Arbor, MI, USA), following the manufacturer’s instructions. MDA levels were determined according to a previous report [20].

### 4.8. TdT-Mediated dUTP Nick-End Labelling (TUNEL) Assay

DNA fragmentation induced by HG was examined using the In Situ Cell Death Detection Kit, Fluorescein (Roche Diagnostics GmbH, Mannheim, Germany), according to the manufacturer’s instructions. In brief, cells cultured under experimental conditions in a 9.5 mm × 4 well Multi-Well Glass Bottom Dish were fixed with 4% paraformaldehyde phosphate-buffered solution and then permeabilized with 0.1% trisodium citrate and 0.1% Triton X-100 in PBS. The cells were incubated with a TUNEL reaction mixture for 1 h at 37 °C, followed by the addition of 0.2% DAPI solution for nuclear staining. The stained cells were examined using a BIOREVO BZ-9000 microscope (KEYENCE CORPORATION., Osaka, Japan). Cells were considered apoptotic when they exhibited green fluorescence (TUNEL-positive cells) when visualized by the incorporation of labeled deoxyuridine triphosphate (dUTP) at the sites of DNA breaks in a reaction catalyzed by the deoxynucleotidyl transferase (TdT) enzyme. ImageJ software version 1.52a (National Institutes of Health, MD, USA) was used to determine the number of TUNEL-positive cells (green fluorescence) by labeled dUTP and the number of nucleus-stained cells (blue fluorescence) by DAPI solution.

### 4.9. Statistical Analysis

All data are expressed as the mean ± S. E. M. Statistical significance was evaluated using one-way analysis of variance (ANOVA), followed by the Bonferroni test. Statistical significance was set at *p* < 0.05.

## Figures and Tables

**Figure 1 ijms-23-05187-f001:**
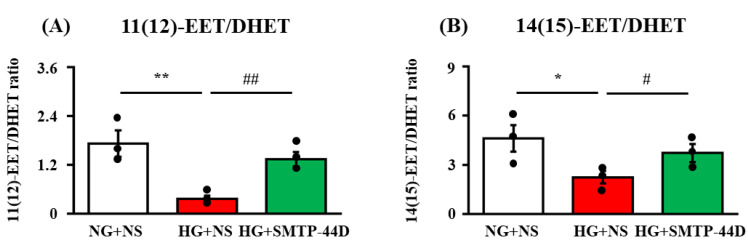
The ratio of 11(12)-EET/DHET (**A**) and 14(15)-EET/DHET (**B**) in response to SMTP-44D in IMS32 cells under high glucose conditions. IMS32 cells were incubated with SMTP-44D (30 μM) from 48 to 96 h after the HG treatment. The levels of EETs and DHETs were determined 96 h after the HG treatment. The data are expressed as the mean ± S. E. M. (*n* = 3). * *p* < 0.05, ** *p* < 0.01 vs. NG + NS group; #*p* < 0.05, ##*p* < 0.01 vs. HG + NS group by one-way analysis of variance followed by Bonferroni test. NG, normal glucose; HG, high glucose; NS, normal saline.

**Figure 2 ijms-23-05187-f002:**
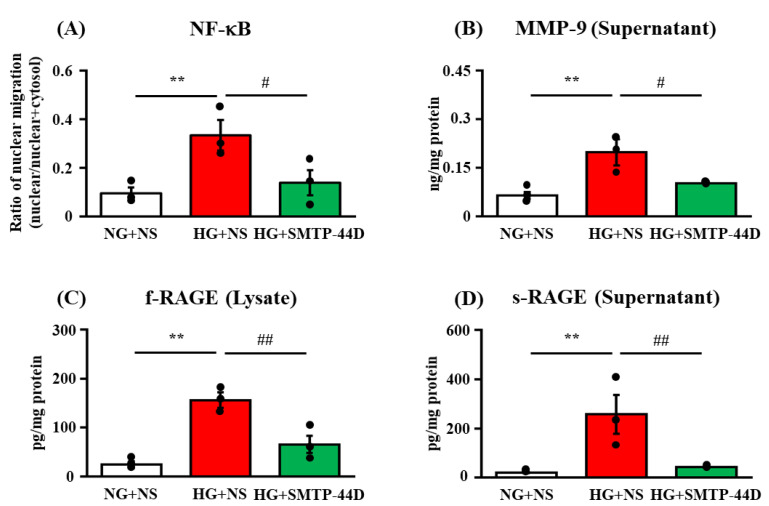
Nuclear migration of NF-κB (**A**) and the levels of MMP-9 (**B**), f-RAGE (**C**), and s-RAGE (**D**) in response to SMTP-44D in IMS32 cells under high glucose conditions. IMS32 cells were incubated with SMTP-44D (30 μM) from 48 to 96 h after the HG treatment. The nuclear migration of NF-κB (**A**) and the levels of MMP-9 (**B**), f-RAGE (**C**), and s-RAGE (**D**) were determined 96 h after the HG treatment. The data are expressed as the mean ± S. E. M. (*n* = 3). ** *p* < 0.01 vs. NG + NS group; #*p* < 0.05, ##*p* < 0.01 vs. HG + NS group by one-way analysis of variance followed by Bonferroni test. NG, normal glucose; HG, high glucose; NS, normal saline.

**Figure 3 ijms-23-05187-f003:**
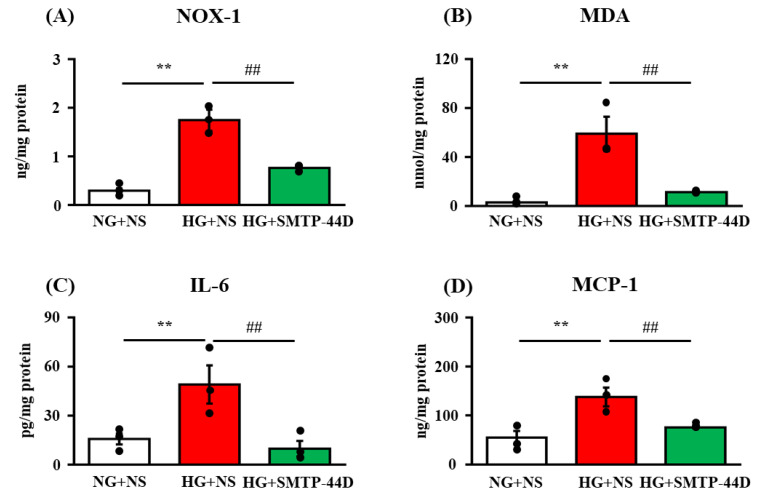
The levels of NOX-1 (**A**), MDA (**B**), IL-6 (**C**), and MCP-1 (**D**) in response to SMTP-44D in IMS32 cells under high glucose conditions. IMS32 cells were incubated with SMTP-44D (30 μM) from 48 to 96 h after the HG treatment. The levels of NOX-1 (**A**), MDA (**B**), IL-6 (**C**), and MCP-1 (**D**) were determined 96 h after the HG treatment. The data are expressed as the mean ± S. E. M. (*n* = 3). ** *p* < 0.01 vs. NG + NS group; ##*p* < 0.01 vs. HG + NS group by one-way analysis of variance followed by Bonferroni test. NG, normal glucose; HG, high glucose; NS, normal saline.

**Figure 4 ijms-23-05187-f004:**
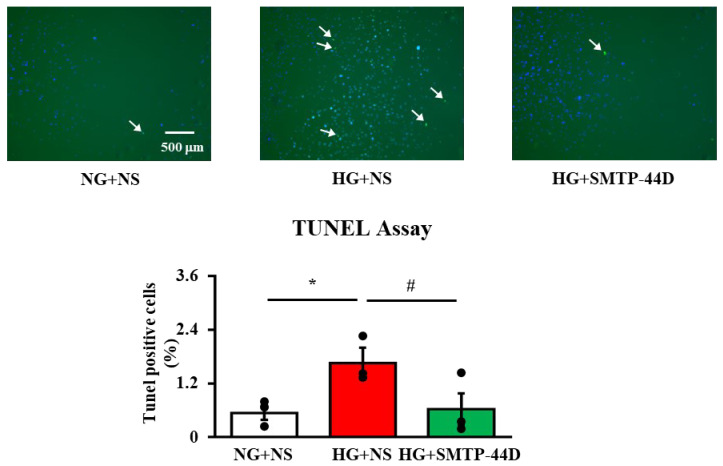
Evaluation of apoptosis in response to SMTP-44D in IMS32 cells under high glucose conditions. IMS32 cells were incubated with SMTP-44D (30 μM) from 48 to 96 h after the HG treatment. Apoptosis analysis by the TUNEL assay was performed 96 h after the HG treatment. Representative images of TUNEL-positive cells (green fluorescence) labeled with dUTP and nuclei (blue fluorescence) stained with DAPI solution are shown. White arrows indicate TUNEL-positive cells. The data are expressed as the mean ± S. E. M. (*n* = 3). * *p* < 0.05 vs. NG + NS group; #*p* < 0.05 vs. HG + NS group by one-way analysis of variance followed by Bonferroni test. NG, normal glucose; HG, high glucose; NS, normal saline.

**Figure 5 ijms-23-05187-f005:**
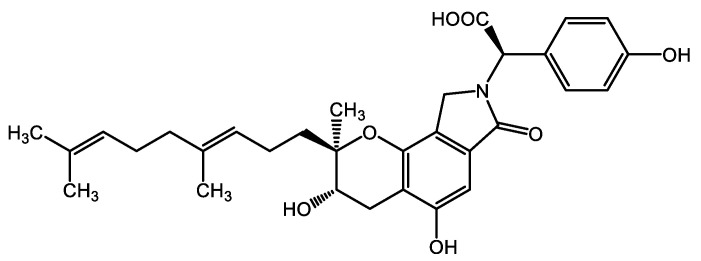
Chemical structure of SMTP-44D.

**Figure 6 ijms-23-05187-f006:**
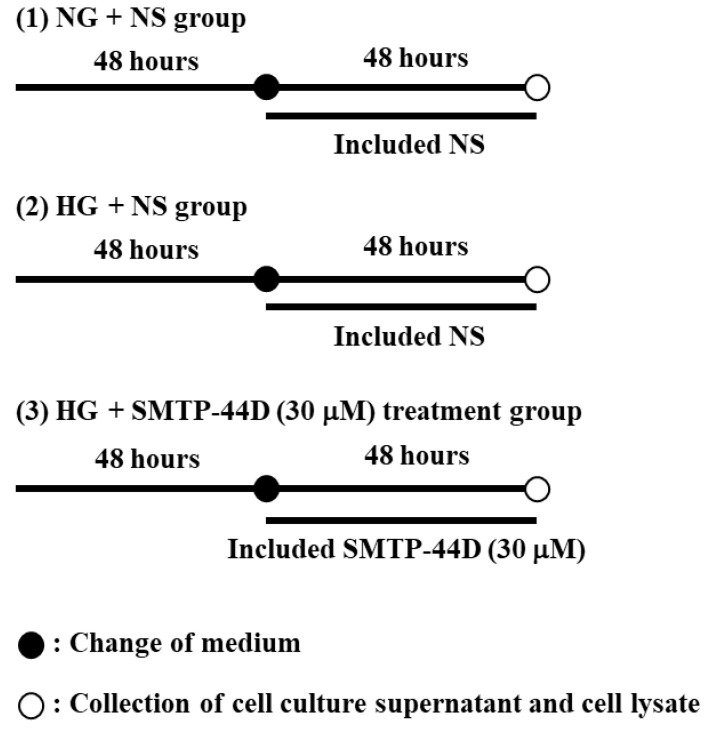
Schematic representation of the three different culture conditions: (1) Normal glucose (5.6 mM; NG) + normal saline (NS); (2) high glucose (30 mM; HG) + NS; (3) HG + SMTP-44D (30 μM).

## Data Availability

The data presented in this study are available in this article.

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
