# Peer review of "SMTP-44D Exerts Antioxidant and Anti-Inflammatory Effects through Its Soluble Epoxide Hydrolase Inhibitory Action in Immortalized Mouse Schwann Cells upon High Glucose Treatment"

_ijms, 2022, doi:10.3390/ijms23095187_

Round 1

Reviewer 1 Report

Omega-3 and omega-6 fatty acids can be converted into bioactive epoxides by cytochrome P450s (CYP450), which play pro-resolving roles in the inflammatory response. However, soluble epoxide hydrolase (sEH) converts epoxides into diols, which lack pro-resolving functions and could be cytotoxic.

The diabetic milieu is hyperglycemic, but there are also increased fatty acids. So, I think that it would be most appropriate if the cells had been incubated with omega-3 and omega-6 fatty acids. In addition, hypertriglyceridemia has been considered a significant risk factor for diabetic neuropathy, and maybe increased triglycerides could have been involved in the pathogenesis of neuropathy. Therefore, the experiment would be much more intriguing.

Moreover, the authors should detail the suggested common metabolic pathway between fatty acids, polyol, and the formation of ages.

The methodology is appropriate for current research, but the authors could improve it in future studies. However, the manuscript is clearly written, and the discussion/conclusions are acceptable.

Overall, data are of interest.

Author Response

Answer to the comments from Reviewer: 1

Comments to the Author

  1. Omega-3 and omega-6 fatty acids can be converted into bioactive epoxides by cytochrome P450s (CYP450), which play pro-resolving roles in the inflammatory response. However, soluble epoxide hydrolase (sEH) converts epoxides into diols, which lack pro-resolving functions and could be cytotoxic.

>> 

Thank you very much for your valuable comments. We realize that it is very important to evaluate the toxicity of diols. We would like to evaluate the toxicity of diols as it will be an important factor in future studies on diabetic neuropathy.

  1. The diabetic milieu is hyperglycemic, but there are also increased fatty acids. So, I think that it would be most appropriate if the cells had been incubated with omega-3 and omega-6 fatty acids. In addition, hypertriglyceridemia has been considered a significant risk factor for diabetic neuropathy, and maybe increased triglycerides could have been involved in the pathogenesis of neuropathy. Therefore, the experiment would be much more intriguing. 

>> 

Thank you very much for your interesting comments. As you have mentioned, if the IMS32 cells had been incubated with omega-3 and omega-6 fatty acids, it would be most appropriate as a pathological condition for diabetic neuropathy. However, we would like to clarify how the antioxidant and anti-inflammatory activities of SMTP-44D by hyperglycemia are shown, so in the present study, we studied it under hyperglycemia-only conditions.

  1. Moreover, the authors should detail the suggested common metabolic pathway between fatty acids, polyol, and the formation of ages.

>> 

Based on your suggestion, we have added the sentences as shown below.

(Introduction, line 7, page 4)

 The persistence of hyperglycemia induces oxidative stress and fatty acid increases, which activate the DN-associated metabolic pathways, namely the protein kinase C, polyol, advanced glycation end products, and hexosamine pathways. Especially, oxidative stress is considered the final common pathway of cellular injury under hyperglycemic conditions [3].

  1. The methodology is appropriate for current research, but the authors could improve it in future studies. However, the manuscript is clearly written, and the discussion/conclusions are acceptable.

>> 

Thank you very much for your valuable comments. As you have mentioned, we would like to improve the methodology in future studies.

  1. Overall, data are interest.

>> 

Thank you very much for your interest in our research.

Thank you very much for your valuable comments and suggestions.

Reviewer 2 Report

DN is one of the most common complications of diabetes mellitus. Following high glucose treatment in immortalized mouse Schwann cells (IMS32), the authors aimed to determine how SMTP-44D inhibits soluble epoxide hydrolase (sEH).  Cells (IMS32) were incubated in high glucose medium for 48 hours and then treated with SMTP-44D for 48 hours.  Authors have measured oxidative stress markers, such as NADPH-oxidase-1 and malondialdehyde, inflammatory factors, such as the ratio of nuclear to cytosolic levels of NF-kB, and levels of IL-6, MCP-1, MMP-9, receptor for advanced glycation end product (RAGE), and apoptosis after incubation only using ELISA and MS.  A high glucose treatment with SMTP-44D resulted in a higher ratio of EETs to DHETs and slowed oxidative stress, inflammation, RAGE induction, and apoptosis. The results suggest that SMTP-44D can suppress the induction of apoptosis via anti-oxidant and anti-inflammatory properties, perhaps through inhibition of sEH.  Overall, this reviewer found the article interesting to read. However, I have a few concerns. 1. Translocation of Nf-KB and levels of IL-6, MCP-1, MMP-9 and RAGE can be displayed using immunostaining (Nf-KB) and western blot (IL-6, MCP-1, MMP-9 and RAGE), as measuring via ELISA likely will prevent the readers from observing a clear difference between the experimental groups. 2. Do the authors expect the same effect of SMTP-44D in other cell types such as oligodendrocytes, astrocytes and neurons? Authors could demonstrate this experimentally. 3. Does SMTP-44D cross the blood-nerve barrier? 4. Authors could present all the bar graphs with the visible scattered individual data points.

Author Response

Answer to the comments from Reviewer: 2

Comments to the Author

  1. Translocation of Nf-KB and levels of IL-6, MCP-1, MMP-9 and RAGE can be displayed using immunostaining (Nf-KB) and western blot (IL-6, MCP-1, MMP-9 and RAGE), as measuring via ELISA likely will prevent the readers from observing a clear difference between the experimental groups.

>> 

Thank you very much for your valuable comments. As you have mentioned, measuring via ELISA compared to immunostaining or western blotting likely will prevent the readers from observing a clear difference between the experimental groups. However, to clarify the difference between the experimental groups, NF-κB was quantified as the ratio of nuclear fraction and nuclear fraction + cytoplasmic fraction, while IL-6, MCP-1, MMP-9, and RAGE were quantified as total protein contents. But in future studies, immunostaining and western blotting should be performed to observe a clear difference between the experimental groups.

  1. Do the authors expect the same effect of SMTP-44D in other cell types such as oligodendrocytes, astrocytes and neurons? Authors could demonstrate this experimentally.

>> 

Thank you very much for your interesting comments. We expect that SMTP-44D would have the same effects in other cell types, such as oligodendrocytes, astrocytes, and neurons. In future studies, we would like to experiment with other cell types.

  1. Does SMTP-44D cross the blood-nerve barrier?

>> 

Thank you very much for your valuable comments. At present, we do not have enough data that support the penetration of SMTP-44D into nerve cells in vivo. In our previous study using a diabetic neuropathy mouse model, we demonstrated that SMTP-44D significantly reduced the levels of inflammatory cytokines such as TNF-α, IL-1β, and IL-6 as well as the level of malondialdehyde, an index of oxidative stress in the sciatic nerve (Shinouchi et al., Pharmacol Res Perspect, 2020). These results are consistent with the data obtained in this study using the mouse Schwann cell line IMS32. Thus, SMTP-44D has an activity to improve nerve cell inflammation directly, whereas these facts do not exclude the possibility that SMTP-44D indirectly acts on nerve cells in vivo, and physicochemical evidence of the distribution of the agent in nerve cells is required to reach a definite conclusion.

  1. Authors could present all the bar graphs with the visible scattered individual data points.

>> 

Based on your comment, we have revised Figures with changes like the visible scattered individual data points in all the bar graphs (Figures, pages 35 to 38).

Thank you very much for your valuable comments and suggestions.